# The 4 youth by youth HIV self-testing crowdsourcing contest: A qualitative evaluation

**Juliet Iwelunmor**[1]*, **Oliver Ezechi**[2], **Chisom Obiezu-Umeh**[1], **Titilola Gbaja-Biamila**[2], **Ucheoma Nwaozuru**[1], **David Oladele**[2], **Adesola Z. Musa**[2], **Ifeoma Idigbe**[2], **Florida Uzoaru**[1], **Collins Airhihenbuwa**[3], **Kathryn Muessig**[4], **Donaldson F. Conserve**[5], **Bill Kapogiannis**[6], **Joseph D. Tucker**[7,8]

**1** College for Public Health & Social Justice, Saint Louis University, Saint Louis, Missouri, United States of America, **2** Clinical Sciences Department, Nigerian Institute of Medical Research, Lagos, Nigeria, **3** Heath Policy and Behavioral Sciences, School of Public Health, Georgia State University, Atlanta, Georgia, United States of America, **4** Gillings School of Global Public Health, University of North Carolina at Chapel Hill, Chapel Hill, North Carolina, United States of America, **5** Department of Health Promotion, Education, and Behavior, Arnold School of Public Health, University of South Carolina, Columbia, South Carolina, United States of America, **6** Maternal and Pediatric Infectious Diseases Branch, Eunice Kennedy Shriver National Institute of Child Health and Human Development, National Institutes of Health, Bethesda, Maryland, United States of America, **7** Division of Infectious Diseases, School of Medicine, University of North Carolina at Chapel Hill, North Carolina, United States of America, **8** Faculty of Infectious and Tropical Diseases, London School of Hygiene and Tropical Medicine, London, United Kingdom

* juliet.iwelunmor@slu.edu

## Abstract

### Introduction

Crowdsourcing, a participatory approach to solicit ideas from a large group of diverse individuals, provides an opportunity to nurture youth participation in HIV self-testing service design. The objective of this study was to describe the responses to a crowdsourcing contest aimed at soliciting ideas on how to promote HIV self-testing (HIVST) among young people in Nigeria.

### Methods

The "4 Youth by Youth" HIV Self-Testing Crowdsourcing contest was an online and off-line contest that asked Nigerian youth (10–24 years old) for ideas in response to the following: How might we promote HIVST among young people in Nigeria? All data were collected and analyzed between October 2018, and June 2019. Ideas and perceptions generated from the crowdsourcing contest were qualitatively analyzed using thematic content analysis. Specifically, four reviewers analyzed whether the ideas generated were desirable (appealing to young people), feasible (easy to implement) and impactful (will significantly influence HIVST uptake among young people).

### Results

A total of 903 entries were received in response to the contest prompt. Participants submitted entries in various forms: online form (39.7%), offline Dropbox (44.6%), email (6.1%) and

**Data Availability Statement:** Il relevant data are within the manuscript.

**Funding:** This study was funded by Eunice Kennedy Shriver National Institute of Child Health

and Human Development Grant number: 1UG3HD096929 and NIAID K24AI143471

**Competing interests:** All authors declare that they have no competing interest.

WhatsApp (9.7%). Of the total entries, 85% (n = 769/903) entries were eligible and were scored as having either high, moderate or low level of feasibility, impact and desirability, on a 3-point Likert scale. A significant portion of the entries were given a score of 3 for feasibility (4.9%), desirability (7.1%), impact (3.0%) or a total overall score of 7 or more (8.2%). The three main themes that emerged from the entries include:1) Peer-to-peer distribution and leveraging on existing infrastructures 2) Youth-Oriented Branding of the HIVST Kit 3) Mobile platforms and social media technology.

## Conclusion

The "4 Youth by Youth" Self-Testing contest engaged a broad audience of young people to generate ideas and perspectives on how to promote HIVST. This process informed the development of youth innovated implementation strategies to increase uptake of HIVST among adolescents and youth at risk for HIV.

## Introduction

Despite the advances made in the HIV research arena, including several groundbreaking milestones in biomedical prevention of HIV infection, the rate of new HIV infections among youth at risk has steadily increased over the past decade [1]. Nigerian youth are at the epicenter of an expanding HIV crisis, with the second largest number of new youth HIV infections of any country and nearly one in five youth aged 15–24 have never tested for HIV [2]. Individual (i.e. low perceptions of risk) [3, 4], social (gender norms) [5], and structural (i.e. costs, stigma) barriers [2–4, 6] remain serious challenges to the delivery of youth-friendly HIV prevention services, including HIV testing for young people in Nigeria [7].

HIV self-testing (HIVST) can mitigate some of these barriers by decentralizing spaces for HIV testing as individuals collect their own specimen and receive results in privacy [8–12] thus decreasing stigma. HIVST has shown promise for linking HIV-negative youth to prevention services and HIV-positive youth to treatment and care [13–16]. Thus, more young people who may not otherwise test are empowered to do so discreetly and conveniently [12, 17, 18]. The World Health Organization [9] and the Nigerian National AIDS Strategic Framework [19] recommend HIVST as an additional approach to delivering HIV testing services. HIVST benefits include convenience, simplicity, privacy, confidentiality, and autonomy in controlling one's health [14, 15, 18, 20–22]. However, there are few HIVST programs that are focused on serving young people [14, 23, 24] or organized by youth themselves [25, 26] and none have been neither led by youth nor rigorously evaluated to see if they improve HIV testing rates in youth.

Crowdsourcing, the process of shifting a task to a large group of people in the form of a contest or open call and then sharing solutions with the public, may be useful to create HIVST services tailored for young people [27–30]. One major benefit of crowdsourcing is that the solutions are often appropriate and highly relevant to the intended audience or key stakeholders who themselves are directly involved in creating and/or implementing the proposed solutions [28, 31, 32]. Prior research has used crowdsourcing to determine youth preferences for asthma self-management [33], multilingual health promotion messages for oral health [34], peer-generated messages for alcohol prevention programs [35], and to solicit videos to promote HIV testing [27, 36].

We launched an open challenge contest to elicit youth perspectives and preferences for promoting HIVST within Nigeria (i.e. what, when, where, or who distributes HIVST). Specifically, we focused on Nigerian youth ages 10–24 as potential end-users of HIVST as they remain at increased risk for acquiring HIV, yet the majority are unaware of their status. The purpose of this study is to examine youth responses to an HIVST crowdsourcing contest in Nigeria to promote the uptake of HIVST among young people.

## Methods

### The contest

A challenge contest was organized and hosted by the 4 Youth by Youth (4YBY) group, which is a team of young people, health professionals, activists, and entrepreneurs from diverse backgrounds, who are united by the shared passion to advance Nigerian youth participation in creating innovative, sustainable HIV services. The contest was called "The World AIDS Day HIV Self-Testing Contest" to mark the annual global December 1st World AIDS Day event in Lagos, Nigeria. The goal of the contest was to solicit ideas and/or concepts in response to the following prompt: How might we promote HIVST among young people in Nigeria? The hashtag "HIVSELFTESTINGCONTEST" identified the campaign on social media platforms. To hold the contest, we organized a multi-sectoral contest advisory panel; engaged youth from diverse backgrounds and cultures to participate; evaluated ideas based on pre-specified criteria; selected the top 30 finalists and invited them to deliver a 3-minute pitch of their idea to a panel of judges on World AIDS Day; and announced finalists and winners by end of Day event. In organizing the contest, we used the TDR Practical Guide on Crowdsourcing in Health and Health Research.[37] This research was approved by the Saint Louis University Ethics Committee and the Nigerian Institute of Medical Research Ethics Review Board.

### Establishment of an advisory panel

A multisectoral contest advisory panel was established with representation from key stakeholders involved with youth and community engagement, including 4YBY youth ambassadors, crowdsourcing experts, HIV prevention researchers, entrepreneurs and communication experts with keen knowledge of the Nigerian context. The contest advisory panel guided the overall development, promotion and evaluation of the contest.

### Participant recruitment and dissemination of the call for entries

We invited all young people between the ages of 10 to 24 years in Nigeria to participate in the HIVST-themed crowdsourcing contest. We utilized purposive sampling techniques to ensure that a range of young people from diverse backgrounds were engaged. From October 1, 2018 to November 16, 2018, an open call was disseminated online on social media sites such as Facebook and Instagram as well as in-person events within secondary schools, universities and community centers where young people congregate. In-person events held to promote the contest included presentations led by contest organizers, with feedback sessions to answer questions. In collaboration with a Nigerian communications company, a short video was developed and posted online on the 4YBY website to promote the contest. In addition, a member of the 4YBY youth ambassadors promoted the contest at five secondary schools in three different local government areas (LGA) in Lagos state and encouraged students in those schools to apply for the HIV open challenge contest. Contest fliers were sent to ten individual schools and 6 educational district officers in Lagos state. Finally, three 4YBY ambassadors

announced to their peers in universities about the contest and placed contest banners at strategic campus locations.

## Contest platform and data collection

The initial selection criteria were that participants had to be between the ages of 10 to 24 years, and residing in Nigeria, and their ideas had to describe novel strategies to promote uptake of HIVST among Nigerian youth in English. Participants submitted demographic details upon submission, including, contact information, age, current location, sex, occupation and, level of education. Submission of entries could be in form of written descriptions (150 words or less), images, drawings, posters, videos, taglines, describing how to promote HIVST among young people in Nigeria. Participants were given the option to submit their ideas either online (via Google online forms, WhatsApp or email) or offline (paper-based version, handwritten or typed). Participants interested in joining the contest completed written informed consent either online or offline. Participants had the opportunity to pose questions to the study team via social media, WhatsApp messaging or Email.

## Evaluation of Ideas

The eligible entries were further reviewed by four contest organizers and rated based on their feasibility, desirability and impact using a 3-point scale (3 = high, 2 = moderate or 1 = low level), which was adapted from the human-centered, design thinking framework [38]. Two members of the research team initially screened all 903 entries received for duplicates and relevance to the scope of the contest. Entries that could not be scored for any of the three criteria, entries from participants above 25 years, and those that were duplicate submissions were excluded from further analysis. Subsequently, each of the four reviewers independently rated the ideas submitted using the following three criteria: 1) desirability- Is the idea appealing to youth? Does the idea meet the needs (low-cost, accessible, confidential) of young people?; 2) feasibility- Can the idea be easy to implement? Are the resources available to execute the idea?; and 3) Impact- Will the idea significantly influence young people to self-test for HIV? Is the idea able to reach young people in Nigeria when available? Finally, the scores for the three criteria were summed to generate an overall score for each of the entries with a range of 3 to 9. Entries that received an overall score greater than 7 (pre-specified by the advisory panel) on a scale of 3–9 were invited to present a 3-minute pitch at the annual World AIDS Day event. Three members of the contest advisory panel and two key community stakeholders served as the judges for the event. The top three finalists, based on cumulative votes by the judges, were selected and awarded prizes.

## Data analysis and coding

After the contest was completed, each participant entry was deidentified and transcribed to allow for a thematic analysis of data [39, 40]. Transcripts were coded by research staff trained in qualitative analysis methods and entered in Microsoft Excel 2016. Descriptive statistics were used to describe participant demographics and characteristics of the submissions in SAS version 9.4. Entries were initially coded by 4 trained research staff (JI, CO, UN and FU) and content analysis was performed to code the data and identify similar patterns or categories. A modifiable coding sheet informed by the data was used to code the entries to identify commonalities and differences between the data. Following initial coding, the entries were reread to refine the coding sheet and a sample of transcripts were provided by two of the researcher staff, external to the study team, who independently reviewed and confirmed themes and coding. Once the themes and sub-themes were finalized, CO and UN conducted the final coding

of the entries. A third researcher confirmed the analysis and resolved any discrepancies that occurred with the coding process. The Consolidated Criteria for reporting qualitative research (COREQ) guided the reporting of the results.

# Results

## Key characteristics and quality of contest entries

A total of 903 entries were submitted by young Nigerians between the ages of 10 to 24 years in response to the open challenge contest call. Of those entries, 60% (n = 550) of the participants submitted their ideas online (website application form = 425; WhatsApp = 78; Email = 47), and the remaining 40% (n = 353) submitted a paper version of the application. Most the entries were submitted by individuals residing in Lagos state (91%, n = 819). However, we shortlisted the entries based on pre-specified criteria. Of the 134 entries (15%) that were not eligible, 6 of those were duplicate entries, 113 were entries that could not be scored and 15 were entrants who were older than 24 years. In total, we found 769 eligible, non-duplicated entries submitted by 769 applicants. Among these youth, 33.6% were less than 14 years, 44.2% were between the ages of 15 to 19 years, and 22.2% were between the ages of 20 to 24 years old. Approximately half of the participants were females (51.2%) and more than half had obtained a primary education (52.6%), whereas, 47.4% of the participants had obtained a secondary education or higher (refer to Table 1).

Regarding the content of the 769 entries, the average word count of the written entries was 180 words and the average video length for entries was 2.14 minutes (range: 0.3–5.34 minutes). Fifty-four entries used a combination of text with images (refer to Fig 1), 6 entries were a combination of texts and videos, and the remaining entries were completely in text format. Thirty five percent provided an overview on HIV and risk factors among young people, while 61% addressed existing barriers to HIV self-testing. The mean overall score of all eligible entries was 4.0/9.0 (standard deviation = 1.5) with 8.2% receiving a high overall score (7–9). On individual criteria, a high score of 3 was received by about 7.2% of the entries rated on desirability, 4.9% rated on feasibility, and 3.0% rated on impact (refer to Table 1). The most relevant themes for promotion of HIVST that emerged from the open challenges are presented below and in Table 2.

## Peer-to-peer distribution and harnessing existing infrastructures

Peer-to-peer distribution were described as potentially feasible strategies for mobilizing HIVST delivery among young Nigerians. Participant entries suggested that supportive interactions among peers as well as discussions by young people with their peers may promote and increase awareness on proper use of the HIVST kits. Youth also provided ideas on using peer educators as role models or brand ambassadors for HIV self-testing kits. They suggested that these peer educators can also train other young people using the train-the-trainer model so that other youth could become engaged in the delivery of HIV self-testing kits to their peers. In addition, schools, religious institutions and community centers serving young people were also described by participant entries, as existing infrastructures that could be potentially harnessed to promote HIVST delivery. Participants suggested that mobilization activities should focus on these infrastructures since they already interact with large numbers of young people. Example of feasible themes are included in Table 2.

## Youth-oriented branding of the HIVST kit

Overall, participant entries included several suggestions for making HIVST more appealing to young Nigerians. This included repackaging existing HIVST products with colors, taglines,

**Table 1. Key characteristics of contest entries, HIVST crowdsourcing contest participants—Nigeria, 2018 (n = 769 eligible entries).**

| | N | % |
|---|---|---|
| **Total** | 769 | 100.0 |
| **Age (years)** | | |
| 10–14 | 139 | 33.6 |
| 15–19 | 183 | 44.2 |
| 20–24 | 92 | 22.2 |
| **Gender** | | |
| Female | 376 | 51.2 |
| Male | 358 | 48.8 |
| **Educational attainment** | | |
| Primary | 271 | 52.6 |
| Secondary | 220 | 42.7 |
| Tertiary | 24 | 4.7 |
| **Mean overall score, (SD)** [†] | 4.0 | (1.5) |
| **Feasibility** | | |
| Low | 526 | 68.4 |
| Moderate | 205 | 26.7 |
| High | 38 | 4.9 |
| **Desirability** | | |
| Low | 486 | 63.2 |
| Moderate | 228 | 29.6 |
| High | 55 | 7.2 |
| **Impact** | | |
| Low | 614 | 79.8 |
| Moderate | 132 | 17.2 |
| High | 23 | 3.0 |

HIVST = HIV Self-Testing, SD = standard deviation.

Some frequencies do not add up to the total due to missing observations.

[†] Four independent judges scored the contest entries based on 3 criteria (feasibility, desirability and impact) on a 1–3 scale (3 = high, 2 = moderate or 1 = low level). Scores for the 3 criteria were summed (total up to 9 points) and the average overall score for all entries was computed.

designs, and animations that are youth-friendly (75%). A significant number of entries highlighted making the package smaller and more flexible in order to be discreet. Additionally, several entries suggested providing instructions translated in the three most common Nigerian languages (Igbo, Hausa and Yoruba) to enhance appeal to a diverse segment of youth in Nigeria. Notably, several participant entries described a comprehensive approach to HIVST promotion, incorporating different youth-friendly health products, including personal hygiene products such as grooming kits for men and sanitary pads for females to enhance HIVST appeal. Finally, participant entries emphasized the need for HIVST products targeting youth to be low cost with price ranging from 500–1000 naira ($1.38-$2.78) so as to be affordable to Nigerian youth. Example of desirable themes are included in Table 2.

## Mobile platforms and social media technology

With the high penetration of mobile technology and social media use among young people in Nigeria, 11.3% of entries suggested the use of these technologies to promote HIVST among

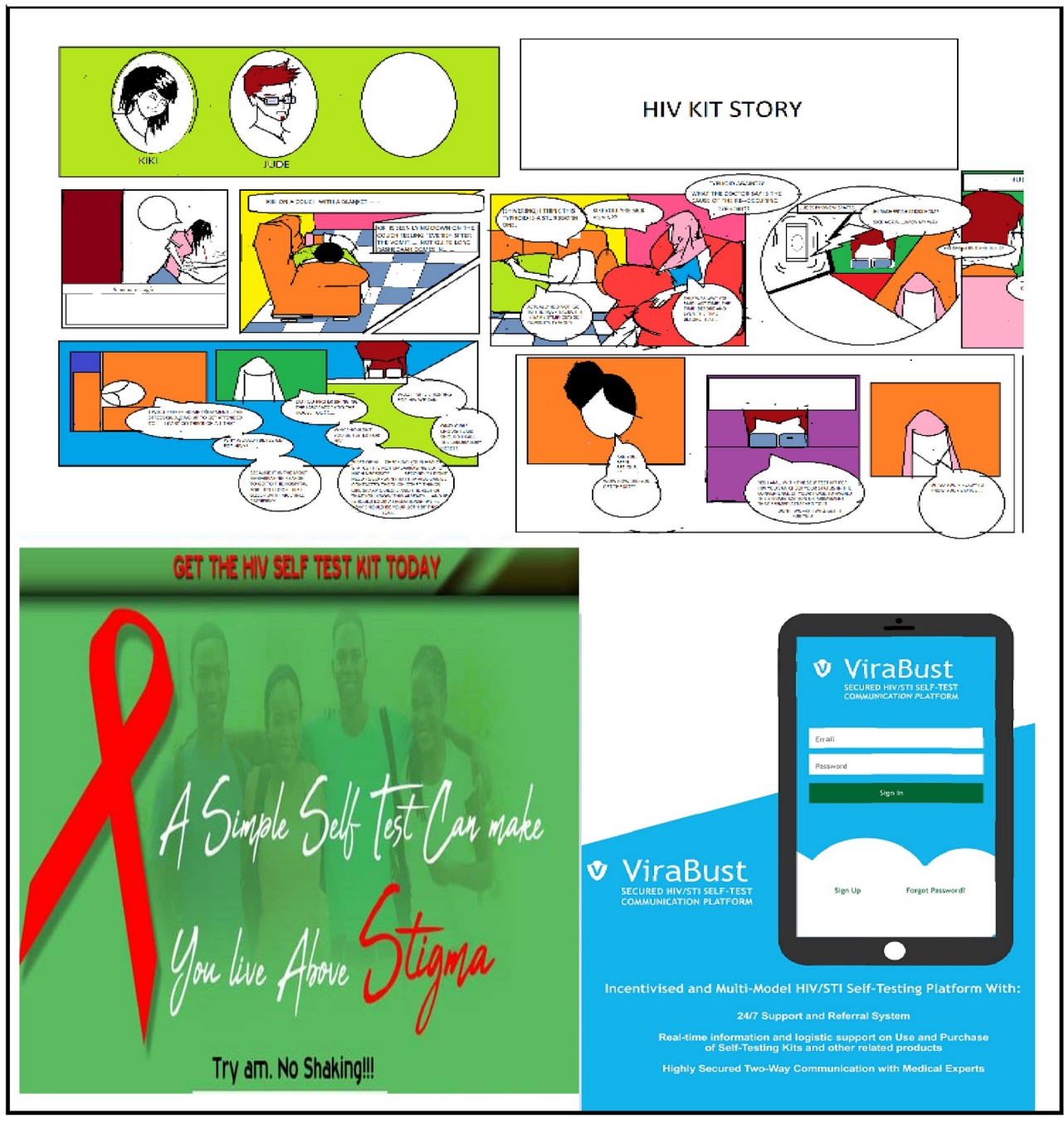

**Fig 1. Sample images submitted to the 4 youth by youth HIV self-testing contest.**

youth in Nigeria. Some of the entries suggested using local context-specific taglines and hashtags to increase demand for HIVST on social media. Similar to popular viral campaigns on social media, participants' entries suggested that these locally derived taglines or hashtags are common in Nigeria and can be used as tools to create awareness and mobilize HIVST delivery to young Nigerians. Entries also suggested recruiting local celebrities to endorse these hashtags to generate demand not only among these celebrities but also among their young fans.

**Table 2. Illustrative themes categorized by the feasibility, desirability and impact of the proposed ideas by 769 young people, ranging from 10 to 24 years in Nigeria.**

| Judging Criteria | HIV Self-testing strategy | Illustrative Themes |
|---|---|---|
| **Feasible ideas for HIVST Delivery** (Can the idea be easy to implement? Are the resources available to execute the idea?) | Peer-to-peer distribution and existing infrastructures | ". . . introducing the HIV self-testing kit to them, train them on how to use, and encourage each of them to have group of ten and train them at community level for a certain period while each of those peers would also have their group to be train at the end of their passing out. . ." (No. O017) |
| | | "Five to ten students will be selected from each school for training including their mentors on contents that will include: Meaning of AIDS, HIV and STIs, what is HIV Testing, . . .Communication Skills. . ., abstinence and other life sustenance skills. . ." (No. O053) |
| | | ". . . Would engage youth as Peer Educators, counsellors and caregivers charged with the responsibility of client follow up, marketing and sales of test kits. . ." (No. O009) |
| | | "HIV self-testing can be improved among youth through collaborating with organizations responsible for community-based program as well as workshop to enable youth participate and be enlightened about the self-testing method." (No. O108) |
| | | "Using outreach programs including community-based programs, that are already in place to connect with the young people in Nigeria. . ." (No. O059) |
| **Desirable ideas for HIVST Delivery** (Is the idea appealing to youth? Does the idea meet the needs (low-cost, accessible, confidential) of young people?) | Youth Branding of the HIVST Kit | "Would package the test kit in sky blue color nylon pack sealed; hence can be folded easily for easy carriage. Make it the size of a medium Bible; with each pack containing the test kit, a direction on how to use and read results. . ." (No. O251) |
| | | "Packaged in portable easy to keep 6-inch by 6-inch purple and pink thick nylon or foil containers with the picture of a confident cheerful, relaxed and good-looking young person on it. . ." (No. O257) |
| | | "it should be packaged in a pink or purple color pack; the size of 2 Gold circle packs of condoms, with information leaflet stating the step by step use and how to read result. It must come with human (young persons) pictures and carefully folded into each pack with background information about the test kit, basic facts about HIV & AIDS and a directory of Youth Friendly Organizations providing ART services with contact phone lines (hot lines) for confidential referral and treatment services. Information on the leaflet will come in Yoruba, Igbo, Hausa, English and Pidgin English languages" (No. O039) |
| | | "It should cost between N500 to N1000. . ." (No. O096) |
| **Impactful ideas for HIVST delivery** (Will the idea significantly influence young people to self-test for HIV? Is the idea able to reach young people in Nigeria when available?) | Mobile platforms and social media technology | ". . .awareness can be created using short videos on how HIV self-testing can be done and this will be sent to different social media platform including Facebook, Twitter, Instagram, WhatsApp and so on. Meanwhile we can attach a reward (which could be in form of data or call unit) for those with highest videos shared or highest viewer or possibly the highest videos liked." (No. O126) |
| | | "To start any movement, a short and effective hashtag like '#shaku-test' could be used, which can be unconsciously picked up by the average Nigerian youth. It can be made more effective by using funny memes and tagging Nigerian celebs that are inclined to advocacy." (No. O020) |
| | | ".a mobile interactive application will be created to guide the user on how to carry out the test and support the user where the result is positive" (No. O003) |
| | | "Virabust proposes a mobile health communication platform (app) for HIV and other STI Self-Test procedures with incentivized and multi-model (visual, text, auditory) channels to promote self-testing among young people in Nigeria." (No. O009) |

Notably, some entries suggested that mobile and social media platforms can be used to carry out nationwide campaigns in the form of contests on how to promote HIVST, similar to the one organized by 4YBY platform. Participants suggested that contests are an impactful approach to reach and engage young people and can be used as a strategy to not only mobilize youth, but also to encourage engagement and subsequent uptake of HIVST. Example of impactful themes are included in Table 2.

## Discussion

In this crowdsourcing study conducted with youth aged 10–24 in Nigeria, we elicited youth perspectives to better understand what attributes (i.e. what, when, where, or who distributes HIVST) are most desirable, feasible and potentially impactful with mobilizing the promotion and delivery of HIVST to Nigerian youth. This study expands the literature by implementing a crowdsourcing contest in an LMIC context, effectively soliciting both broad and deep engagement among youth, and engaging a large number of men.

First, we successfully crowdsourced for ideas for promoting HIVST and engaged a large audience of 769 young people. This contest generated a much higher participation rate when compared to other crowdsourcing challenges and is one of the largest crowdsourcing contest for health.[28, 37, 41] Notably, the contest engaged young men and women across Nigeria, both online and offline. More than 66% of the participants were 15 years or older, while 33.6% were younger than 15 years of age. Although few social media or online interventions are traditionally targeted at young people in low- or middle-income countries [42], this study illustrates the potential to access multiple youth age groups through an open challenge contest using existing online platforms. Given that internet bandwidth is rapidly increasing across many African regions, a crowdsourcing approach may be able to reach farther into rural and other under-served areas.

Second, we identified themes that may be useful for future public health campaigns or initiatives that seek to raise awareness and scale up HIVST uptake among young people in Nigeria. Many of the entries submitted in this contest proposed conventional approaches (for example, using outreach programs including community-based programs, that are already in place to connect with the young people in Nigeria) that are considered feasible to implement. However, few proposed unconventional but impactful and desirable ideas (for example, creating awareness using short videos on how to use a HIVST kit, which will then be shared on several social media platforms and viewers will automatically earn bonus points that can be used to gain rewards) that could potentially draw attention to HIVST and make them more noticeable to youth populations. To our knowledge, this study is the first to elicit ideas from young people about how to make HIVST more youth-friendly and our findings could be used to not only create models for HIVST distribution that are desirable to young people, but also kits that would influence uptake of HIVST, thus increasing the proportion of youth who are aware of their HIV status and facilitating appropriate health-seeking behaviors.

Third, an important finding from our contest was the meaningful engagement of young people in generating input, images, and outreach strategies that may potentially enhance demand for HIVST among young people in Nigeria. Findings generated youth ideas on potential mobilization efforts, including where these activities should focus, who should carry out these activities and what additional services or support tools may be incorporated to increase demand and uptake of HIVST among Nigerian youth. Engaging young people to identify solutions on how to promote HIV self-testing is important in developing HIVST campaigns and distribution messages that will resonate with young people. It ensures that crucial youth perspectives, cultural and linguistic appropriateness are incorporated in mobilization strategies to

promote HIVST uptake among youth in Nigeria. Overall, the entries received from the open challenge contests give voice to young people's lived experiences in Nigeria and could serve as an important step with effectively reaching and engaging young people with HIVST, particularly those who may not otherwise test using conventional strategies.

This study has several limitations. Central among them is that, although we received entries on how to promote HIVST Nigerian youth, most of the entries that did not include information on how to promote HIV self-testing were the offline entries, which indicates a need for better communication on the purpose of the contest both in-person (offline) and online. This is to ensure that all potential participants receive adequate and consistent information on the purpose and the deliverables of the contest. There may have been some selection bias in the entries received. The study sample may not have been representative of the general population of young people between the ages of 14 to 24 years. More than half of the contest entries were received online. Hence, this may have resulted in the inadvertent exclusion of individuals with limited access to internet connectivity or individuals who may have missed the open call period. To minimize the bias, we provided individuals with the option to submit their entries using the paper-based version of the online form, which they were able to submit in a drop-box at a secured location. Since the purpose of this study was to test whether we could use open challenges to generate ideas on how to promote HIV self-testing, future studies may use these findings to design and evaluate youth-friendly campaigns that may increase uptake of HIVST among young Nigerians. Nonetheless, the revised Nigerian National HIV and AIDS strategic framework (2019–2021) [19] calls for scaling up HIV self-testing kits to reach under-served populations with high unmet need for HIV testing. The 4YBY HIV Self-testing crowd-sourcing contest facilitated meaningful youth engagement on images, designs, and relevant themes to reach underserved youth populations who have low HIV testing coverage and remain at ongoing HIV risk. Tapping into the rich wisdom of crowds may lay the groundwork for illuminating youth perspectives on HIVST that could potentially increase HIV testing among Nigerian youth.

## Conclusion

Crowdsourcing requires the active participation of key stakeholders[43, 44], in this case the active participation of youth in creating solutions to increase the uptake of HIV testing among their peers. This strengthens their ability to take ownership of their own health while meeting their unique needs and is central to the sustainability of any health intervention [45]. Building on this opportunity, this form of grassroots activism led by youth themselves can be a cornerstone for achieving an AIDS free generation.

## Acknowledgments

We would like to thank the ITEST team, ID Africa, SESH, PinPont Media, 4 Youth by Youth, Youth Ambassadors and other groups that helped to organize the challenge. We would like to thank the program officers and members of the Prevention and Treatment through a Comprehensive Care Continuum for HIV-affected Adolescents in Resource Constrained Settings (PATC3H) Consortium.

## Author Contributions

**Conceptualization:** Juliet Iwelunmor, Oliver Ezechi, Florida Uzoaru, Collins Airhihenbuwa, Kathryn Muessig, Donaldson F. Conserve, Bill Kapogiannis, Joseph D. Tucker.

**Data curation:** Chisom Obiezu-Umeh, Ucheoma Nwaozuru.

**Formal analysis:** Chisom Obiezu-Umeh, Ucheoma Nwaozuru.

**Funding acquisition:** Bill Kapogiannis.

**Investigation:** Titilola Gbaja-Biamila, David Oladele, Adesola Z. Musa, Ifeoma Idigbe.

**Methodology:** Juliet Iwelunmor, Oliver Ezechi, Joseph D. Tucker.

**Project administration:** Juliet Iwelunmor, Oliver Ezechi, Titilola Gbaja-Biamila, David Oladele, Adesola Z. Musa, Ifeoma Idigbe, Joseph D. Tucker.

**Supervision:** Juliet Iwelunmor, Oliver Ezechi, Joseph D. Tucker.

**Visualization:** Chisom Obiezu-Umeh, Ucheoma Nwaozuru.

**Writing – original draft:** Chisom Obiezu-Umeh, Ucheoma Nwaozuru.

**Writing – review & editing:** Juliet Iwelunmor, Oliver Ezechi, Titilola Gbaja-Biamila, David Oladele, Adesola Z. Musa, Ifeoma Idigbe, Florida Uzoaru, Collins Airhihenbuwa, Kathryn Muessig, Donaldson F. Conserve, Bill Kapogiannis, Joseph D. Tucker.

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
