## [Decision Letter · Decision Letter 0]

10 Dec 2019

PONE-D-19-29605

The 4 Youth by Youth HIV Self-Testing Crowdsourcing Contest: A Qualitative Analysis

PLOS ONE

Dear Dr. Iwelunmor,

Thank you for submitting your manuscript to PLOS ONE. After careful consideration, we feel that it has merit but does not fully meet PLOS ONE’s publication criteria as it currently stands. Therefore, we invite you to submit a revised version of the manuscript that addresses the points raised during the review process.

We would appreciate receiving your revised manuscript by Jan 24 2020 11:59PM. To enhance the reproducibility of your results, we recommend that if applicable you deposit your laboratory protocols in protocols.io, where a protocol can be assigned its own identifier (DOI) such that it can be cited independently in the future. For instructions see: http://journals.plos.org/plosone/s/submission-guidelines#loc-laboratory-protocols

We look forward to receiving your revised manuscript.

Kind regards,

Joseph K.B. Matovu, PhD.

Academic Editor

PLOS ONE

Additional Editor Comments:

Since this is a qualitative paper, the authors should ensure that their manuscript addresses the following aspects: 1) defined objectives or research questions; 2) description of the sampling strategy, including rationale for the recruitment method, participant inclusion/exclusion criteria and the number of participants recruited; 3) detailed reporting of the data collection procedures; 4) data analysis procedures described in sufficient detail to enable replication; 5) a discussion of potential sources of bias; and 6) a discussion of limitations.

The authors should also ensure that the paper is formatted according to the COREQ checklist.

2. We note that Figure 1 in your submission contains copyrighted images.

All PLOS content is published under the Creative Commons Attribution License (CC BY 4.0), which means that the manuscript, images, and Supporting Information files will be freely available online, and any third party is permitted to access, download, copy, distribute, and use these materials in any way, even commercially, with proper attribution. For more information, see our copyright guidelines: http://journals.plos.org/plosone/s/licenses-and-copyright.

5. Your ethics statement must appear in the Methods section of your manuscript. If your ethics statement is written in any section besides the Methods, please move it to the Methods section and delete it from any other section. Please also ensure that your ethics statement is included in your manuscript, as the ethics section of your online submission will not be published alongside your manuscript.

Reviewers' comments:

Reviewer's Responses to Questions

**Comments to the Author**

1. Is the manuscript technically sound, and do the data support the conclusions?

Reviewer #1: Partly

Reviewer #2: Yes

2. Has the statistical analysis been performed appropriately and rigorously? 

Reviewer #1: I Don't Know

Reviewer #2: Yes

3. Have the authors made all data underlying the findings in their manuscript fully available?

Reviewer #1: Yes

Reviewer #2: Yes

4. Is the manuscript presented in an intelligible fashion and written in standard English?

Reviewer #1: Yes

Reviewer #2: Yes

5. Review Comments to the Author

Reviewer #1: An interesting and useful article addressing a significant and serious public and personal health issue across the African continent.

The manuscript would be improved by adoption of the following recommendations:

• Recommend inclusion of a definition of ‘Crowd Sourcing” within the abstract – a simple statement explaining that it is a technique to elicit responses from a large group of people.

Methods

The methods section requires significant clarification, specifically in respect to quantitative aspects versus qualitative aspects of the study which appears to be a mixed methods study. Each aspect of methods should be clearly defined in a logical sequence within the script.

Qualitative methods:

• The study included more than 700 participants which would mean a very largescale task in thematic analysis of transcripts and participants quotes. How was this undertaken, who completed the task, what training did they have etc.

• It is recommended that the authors access the COREQ checklist for reporting qualitative research and ensure that the 32 step process is followed in the construction of the manuscript https://www.equator-network.org/reporting-guidelines/coreq/

• There are some significant omissions in the current script. For example, the participant quotes in Table one have not been allocated a code as per the acceptable standards for qualitative reporting.

• The manuscript should be revised and returned with a completed checklist

Quantitative Methods:

The manuscript refers to both survey processes and statistical reporting for example (Line 240 -243) but lacks detailed description of how these methods were designed, implemented or calculated. Significant work needs to be undertaken to detail all aspects of the study methodology. Authors are encouraged to use guidelines within the equator network for development of mixed methods studies inclusive of the use of on-line survey techniques.

The revised manuscript should be returned with the appropriate checklists as confirmation of research rigor and reporting.

Ethics

Additional details of the St Louis and Nigerian Institute of Medical Research ethics clearance should be included within the body of the manuscript, specifically, clearance date and numbers.

Discussion

Requires much greater depth including:

• Implementation policy and considerations

• Feasibility of scaling up recommendations – how do-able and affordable are the recommendations within the themes, for example, how strong are the Telco networks across areas of high HIV prevalence

• Who are the likely implementation partners

• What is the recommended role of government

Reviewer #2: The manuscript is well written The Title answers What, Who Where and When and clearly highlight what is expected in the body. Method section well written and results supports the findings. The findings also supports the conclusion.

An excellent job done in the manuscript

6. PLOS authors have the option to publish the peer review history of their article (what does this mean?). If published, this will include your full peer review and any attached files.

Reviewer #1: No

Reviewer #2: Yes: CAROLINE OCHOLA DANDE; MSC EPIDEMIOLOGY AND BIOSTATISTICS; KEMRI-FACES

---

## [Author Response · Author response to Decision Letter 0]

5 Feb 2020

Dear Dr. Matovu, 

We appreciate the time and effort that you and the reviewers have dedicated to providing your valuable feedback on the attached manuscript. We have been able to incorporate changes to reflect most of the suggestions provided by the reviewers. Here is a point-by-point response to the reviewers’ comments and concerns (please note that responses to the comments are in BOLD and line numbers are in reference to the clean copy of the manuscript attached): 

Additional Editor Comments:

Since this is a qualitative paper, the authors should ensure that their manuscript addresses the following aspects: 

1) defined objectives or research questions; 

We appreciate the helpful feedback. With regards to the research question and study objectives, this has been revised accordingly to clearly state the research question and in this case, the goal of the crowdsourcing contest. Specifically, we stated the following in Page 6, lines 130 – 131:

“The purpose of this study is to examine youth responses to an HIVST crowdsourcing contest in Nigeria to promote the uptake of HIVST among young people.”

2) Description of the sampling strategy, including rationale for the recruitment method, participant inclusion/exclusion criteria and the number of participants recruited; 

Guided by COREQ, we revised the methods section accordingly to include the sampling technique, recruitment strategy, inclusion criteria and number of entrants. Refer to page 8, lines 159 – 176; number of entrants, refer to pg. 11, lines 232 to 240. 

“We invited all young people between the ages of 10 to 24 years in Nigeria to participate in the HIVST-themed crowdsourcing contest. We utilized purposive sampling techniques to ensure that a range of young people from diverse backgrounds were engaged.”

“A total of 903 entries were submitted by young Nigerians between the ages of 10 to 24 years in response to the open challenge contest call.”

3) detailed reporting of the data collection procedures; 

We have clarified the methods section and included a sub-section on “Contest Platform and Data collection”, Pgs. 8 to 9, lines 178 – 190. 

“The initial selection criteria were that participants had to be between the ages of 10 to 24 years, and residing in Nigeria, and their ideas had to describe novel strategies to promote uptake of HIVST among Nigerian youth in English. Participants submitted demographic details upon submission, including, contact information, age, current location, sex, occupation, level of education and marital status. Submission of entries could be in form of written descriptions (150 words or less), images, drawings, posters, videos, taglines, describing how to promote HIVST among young people in Nigeria.”

4) data analysis procedures described in sufficient detail to enable replication; 

We have revised the data analysis section based on your suggestions. Specifically, in Pgs. 10 to 11, lines 213 – 228, we provide detailed description on the data analysis and coding for this study. 

“After the contest was completed, each participant entry was deidentified and transcribed to allow for a thematic analysis of data. Transcripts were coded by research staff trained in qualitative analysis methods and entered in Microsoft Excel 2016. Descriptive statistics were used to describe participant demographics and characteristics of the submissions in SAS version 9.4.”

5) a discussion of potential sources of bias; and 

Thank you for pointing this out. We have included potential sources of bias in the limitation paragraph. Pgs. 19 to 20, lines 362 – 386. For example, we stated the following: 

“There may have been some selection bias in the entries received. The study sample may not have been representative of the general population of young people between the ages of 14 to 24 years”

6) a discussion of limitations

The study limitations are included in the discussion section. Pgs. 19 to 20, lines 362 – 386.

“There may have been some selection bias in the entries received. The study sample may not have been representative of the general population of young people between the ages of 14 to 24 years. More than half of the contest entries were received online. Hence, this may have resulted in the inadvertent exclusion of individuals with limited access to internet connectivity or individuals who may have missed the open call period. To minimize the bias, we provided individuals with the option to submit their entries using the paper-based version of the online form, which they were able to submit in a drop-box at a secured location”

7) The authors should also ensure that the paper is formatted according to the COREQ checklist

We appreciate the helpful comments and utilized the COREQ to revise the manuscript. 

 We have incorporated the PLOS formatting style throughout the manuscript. 

2. We note that Figure 1 in your submission contains copyrighted images.

All PLOS content is published under the Creative Commons Attribution License (CC BY 4.0), which means that the manuscript, images, and Supporting Information files will be freely available online, and any third party is permitted to access, download, copy, distribute, and use these materials in any way, even commercially, with proper attribution. For more information, see our copyright guidelines: http://journals.plos.org/plosone/s/licenses-and-copyright.

 The appropriate permission has been obtained for the images. 

 The ORCID ID for the corresponding author has been included.

 We have included the caption as suggested.

5. Your ethics statement must appear in the Methods section of your manuscript. If your ethics statement is written in any section besides the Methods, please move it to the Methods section and delete it from any other section. Please also ensure that your ethics statement is included in your manuscript, as the ethics section of your online submission will not be published alongside your manuscript.

We agree with this and have incorporated your suggestion in the methods section.

Review Comments to the Author

Reviewer #1: An interesting and useful article addressing a significant and serious public and personal health issue across the African continent.

The manuscript would be improved by adoption of the following recommendations:

• Recommend inclusion of a definition of ‘Crowd Sourcing” within the abstract – a simple statement explaining that it is a technique to elicit responses from a large group of people.

Thank you for the recommendation. The definition of crowdsourcing was included in the abstract. Pg. 2, line 46 – 47. Specifically, we stated the following: 

Crowdsourcing, a participatory approach to solicit ideas from a large group of diverse individuals, provides an opportunity to nurture youth participation in HIV self-testing service design.

Methods

The methods section requires significant clarification, specifically in respect to quantitative aspects versus qualitative aspects of the study which appears to be a mixed methods study. Each aspect of methods should be clearly defined in a logical sequence within the script.

Thank you very much. We utilized the COREQ checklist to revise the manuscript. 

Qualitative methods:

• The study included more than 700 participants which would mean a very largescale task in thematic analysis of transcripts and participants quotes. How was this undertaken, who completed the task, what training did they have etc.

Thank you for pointing this out. Please note that data analysis occurred over a six-month period with 4 trained qualitative researchers who reviewed each transcript. Also, since these are entries to a crowd-sourcing context and not typical qualitative transcripts, the recommended word limit for each entry received was 150 words, in accordance to the practical guide for crowdsourcing in health and health research (see here: https://apps.who.int/iris/bitstream/handle/10665/273039/TDR-STRA-18.4-eng.pdf). As such, the analysis was easy to manage among the 4 trained qualitative researchers, who have either masters in public health or doctoral degree in public health. All the researchers that carried out the data analysis are trained in qualitative analysis methods and have published qualitative research in peer reviewed journals.

• It is recommended that the authors access the COREQ checklist for reporting qualitative research and ensure that the 32 step process is followed in the construction of the manuscript https://www.equator-network.org/reporting-guidelines/coreq/

Thank you very much. We utilized the COREQ checklist to revise the manuscript.

• There are some significant omissions in the current script. For example, the participant quotes in Table one have not been allocated a code as per the acceptable standards for qualitative reporting.

Thank you for this suggestion. We have included the ID number for each quote in Table 1. Pgs. 12-14, lines 259 to 262.

“HIV self-testing can be improved among youth through collaborating with organizations responsible for community-based program as well as workshop to enable youth participate and be enlightened about the self-testing method.” (No. O108)”

• The manuscript should be revised and returned with a completed checklist

Thank you very much. We utilized the COREQ checklist to revise the manuscript.

Quantitative Methods:

The manuscript refers to both survey processes and statistical reporting for example (Line 240 -243) but lacks detailed description of how these methods were designed, implemented or calculated. Significant work needs to be undertaken to detail all aspects of the study methodology. Authors are encouraged to use guidelines within the equator network for development of mixed methods studies inclusive of the use of on-line survey techniques.

The revised manuscript should be returned with the appropriate checklists as confirmation of research rigor and reporting.

We revised the data analysis section to include “Descriptive statistics were used to describe participant demographics and characteristics of the submissions in SAS version 9.4” on Pg 10, lines 217 to 218. In addition, we included how the quantitative data were collected and what variables were included in the section on “contest platform and data collection” on page 8 to 9, lines 181 to 190. 

Ethics

Additional details of the St Louis and Nigerian Institute of Medical Research ethics clearance should be included within the body of the manuscript, specifically, clearance date and numbers.

We included your suggestion on pg. 7, lines 149 to 150.

Discussion

Requires much greater depth including:

• Implementation policy and considerations

Thank you for the suggestion. The in-country policy highlighting HIVST implementation was cited in the manuscript, Pg. 19, lines 378 to 379. Specifically, we stated the following: 

“… revised Nigerian National HIV and AIDS strategic framework (2019-2021) [19] calls for scaling up HIV self-testing kits to reach underserved populations with high unmet need for HIV testing.”

• Feasibility of scaling up recommendations – how do-able and affordable are the recommendations within the themes, for example, how strong are the Telco networks across areas of high HIV prevalence

• Who are the likely implementation partners

• What is the recommended role of government

Thank you for this suggestion. It would have been interesting to explore this aspect. However, in the case of our study, it seems out of scope because the study aim was to examine the responses from young Nigerians to an HIVST crowdsourcing contest and to test whether we could use open challenges to generate ideas on how to promote HIV self-testing.

Reviewer #2: The manuscript is well written The Title answers What, Who Where and When and clearly highlight what is expected in the body. Method section well written and results supports the findings. The findings also supports the conclusion.

An excellent job done in the manuscript

We appreciate the encouraging comments.

---

## [Decision Letter · Decision Letter 1]

30 Mar 2020

PONE-D-19-29605R1

The 4 Youth by Youth HIV Self-Testing Crowdsourcing Contest: A Qualitative Evaluation

PLOS ONE

Dear Dr Juliet Iwelunmor

Thank you for submitting your manuscript to PLOS ONE. After careful consideration, we feel that it has merit but does not fully meet PLOS ONE’s publication criteria as it currently stands. Therefore, we invite you to submit a revised version of the manuscript that addresses the points raised during the review process.

We would appreciate receiving your revised manuscript by April 29, 2020. To enhance the reproducibility of your results, we recommend that if applicable you deposit your laboratory protocols in protocols.io, where a protocol can be assigned its own identifier (DOI) such that it can be cited independently in the future. For instructions see: http://journals.plos.org/plosone/s/submission-guidelines#loc-laboratory-protocols

We look forward to receiving your revised manuscript.

Kind regards,

Joseph K.B. Matovu, PhD.

Academic Editor

PLOS ONE

Additional Editor Comments (if provided):

I agree that the authors should address the methodological issues identified by one of the reviewers (details below).

Reviewers' comments:

Reviewer's Responses to Questions

**Comments to the Author**

1. If the authors have adequately addressed your comments raised in a previous round of review and you feel that this manuscript is now acceptable for publication, you may indicate that here to bypass the “Comments to the Author” section, enter your conflict of interest statement in the “Confidential to Editor” section, and submit your "Accept" recommendation.

Reviewer #1: (No Response)

Reviewer #2: All comments have been addressed

2. Is the manuscript technically sound, and do the data support the conclusions?

Reviewer #1: (No Response)

Reviewer #2: Yes

3. Has the statistical analysis been performed appropriately and rigorously? 

Reviewer #1: No

Reviewer #2: Yes

4. Have the authors made all data underlying the findings in their manuscript fully available?

Reviewer #1: No

Reviewer #2: Yes

5. Is the manuscript presented in an intelligible fashion and written in standard English?

Reviewer #1: Yes

Reviewer #2: Yes

6. Review Comments to the Author

Reviewer #1: Authors have addressed a number of recommendations made in the initial review and the manuscript is much improved, however, some issues remain unresolved.

This study is clearly a mixed method study involving both on-line survey and analysis of qualitative comments within the survey.

Authors were urged to access the Equator Network and submit completed checklists for both qualitative research and on-line surveys. This has not been completed. While there is now evidence of application of several criteria from the COREQ checklist, a number of items remain missing and information is missing in relation to guidelines for reporting findings from on-line surveys. Please do check every item on these checklists.

Information about participants has been expanded but the manuscript lacks of table which would more readily display participant demographics.

Quantitative analysis should be undertaken across the 769 eligible responses. A small number of qualitative comments a recorded against each theme but there is insufficient indication of how many of the 769 participant supported each theme.

Information has now been included about ethics clearance but the manuscript lacks the clearance date and number via each of the committees.

As highlighted this study is clearly a mixed method study involving both on-line survey and analysis of qualitative comments within the survey. Reporting cannot include small scale qualitative analysis alone. Quantitative analysis of the survey responses is needed with the qualitative data then illustrating the themes identified.

Reviewer #2: The manuscript is well written and with all the previous concerns addressed. the introduction/ background, method, results and conclusion are well written and meets the requirements

7. PLOS authors have the option to publish the peer review history of their article (what does this mean?). If published, this will include your full peer review and any attached files.

Reviewer #1: No

Reviewer #2: Yes: Caroline Ochola Dande, Kemri-Faces Program, Kisumu Kenya; cdande@kemri-ucsf.org

---

## [Author Response · Author response to Decision Letter 1]

15 Apr 2020

Dear Dr. Matovu, 

We appreciate the time and effort that you and the reviewers have dedicated to providing your valuable feedback on the attached manuscript. We have been able to incorporate changes to reflect most of the suggestions provided by the reviewers. Here is a point-by-point response to the reviewers’ comments and concerns (please note that responses to the comments are in BOLD): 

Review Comments to the Author

Reviewer #1: Authors have addressed a number of recommendations made in the initial review and the manuscript is much improved, however, some issues remain unresolved. This study is clearly a mixed method study involving both on-line survey and analysis of qualitative comments within the survey. Authors were urged to access the Equator Network and submit completed checklists for both qualitative research and on-line surveys. This has not been completed. While there is now evidence of application of several criteria from the COREQ checklist, a number of items remain missing and information is missing in relation to guidelines for reporting findings from on-line surveys. Please do check every item on these checklists. 

In summary, participants were asked to submit a creative contribution, for example, short descriptions or images that reflected a thoughtful response to the prompt, ‘How will you promote HIV self-testing among young people in Nigeria?’ Participants also completed a demographic section on the submission form (which was available both online and offline/paper-based version). The purpose of this paper was to describe the methodology and key findings from the range of submissions. Similar to Merchant et al. 2014 analysis of crowdsourcing contest entries, themes were identified from the contest submissions and summary statistics were used to describe demographic data. Although we agree that the crowdsourcing contest employs a mixed-methods approach, a thorough discussion on findings from the quantitative analysis is beyond the scope of this paper. We believe that we utilized the appropriate checklist to revise the manuscript. 

Merchant RM, Griffis HM, Ha YP, et al. Hidden in plain sight: a crowdsourced public art contest to make automated external defibrillators more visible. Am J Public Health. 2014;104(12):2306–2312. doi:10.2105/AJPH.2014.302211

Information about participants has been expanded but the manuscript lacks of table which would more readily display participant demographics. Quantitative analysis should be undertaken across the 769 eligible responses. 

Thank you for the suggestion. We agree that the manuscript would benefit from reporting on the descriptive statistics of key characteristics of the contest entries. As a result, we included a table (refer to table 1) to describe participant demographics and key characteristics of the contest entries. 

A small number of qualitative comments a recorded against each theme but there is insufficient indication of how many of the 769 participant supported each theme. Information has now been included about ethics clearance but the manuscript lacks the clearance date and number via each of the committees. 

We identified salient themes throughout the contest submissions and the themes were further characterized by sample representative quotes (refer to table 2). 

As highlighted this study is clearly a mixed method study involving both on-line survey and analysis of qualitative comments within the survey. Reporting cannot include small scale qualitative analysis alone. Quantitative analysis of the survey responses is needed with the qualitative data then illustrating the themes identified.

Thank you for this suggestion. A thorough discussion on findings from the quantitative analysis is beyond the scope of this paper, as the primary focus is to examine the responses to crowdsourcing contest prompt “how will you promote HIV self-testing among young people in Nigeria” and describe key findings from the range of submissions. Reporting on the findings from the quantitative analysis of the responses from the crowdsourcing contest are underway elsewhere. However, we agree that the manuscript would benefit from reporting on the descriptive statistics of key characteristics of the contest entries, as illustrated in table 1. 

Reviewer #2: The manuscript is well written and with all the previous concerns addressed. the introduction/ background, method, results and conclusion are well written and meets the requirements 7. 

We appreciate the encouraging comments.

---

## [Decision Letter · Decision Letter 2]

12 May 2020

The 4 Youth by Youth HIV Self-Testing Crowdsourcing Contest: A Qualitative Evaluation

PONE-D-19-29605R2

Dear Dr. Iwelunmor:

We are pleased to inform you that your manuscript has been judged scientifically suitable for publication and will be formally accepted for publication once it complies with all outstanding technical requirements.

With kind regards,

Joseph K.B. Matovu, PhD.

Academic Editor

PLOS ONE

Additional Editor Comments (optional):

The authors should check the references to ensure that they are well aligned to the journal style. In general, please ensure that all journal names have been presented in standard abbreviated formats, and endeavor to provide weblinks and access dates where documents were obtained from online sources.

Reviewers' comments:

Reviewer's Responses to Questions

**Comments to the Author**

1. If the authors have adequately addressed your comments raised in a previous round of review and you feel that this manuscript is now acceptable for publication, you may indicate that here to bypass the “Comments to the Author” section, enter your conflict of interest statement in the “Confidential to Editor” section, and submit your "Accept" recommendation.

Reviewer #1: All comments have been addressed

2. Is the manuscript technically sound, and do the data support the conclusions?

Reviewer #1: (No Response)

3. Has the statistical analysis been performed appropriately and rigorously? 

Reviewer #1: Yes

4. Have the authors made all data underlying the findings in their manuscript fully available?

Reviewer #1: Yes

5. Is the manuscript presented in an intelligible fashion and written in standard English?

Reviewer #1: Yes

6. Review Comments to the Author

Reviewer #1: This manuscript has been through two previous review cycles and is now much improved. However, more could have been made of the qualitative data and for future publications authors should undertake education to gain more competence in qualitative reporting.

7. PLOS authors have the option to publish the peer review history of their article (what does this mean?). If published, this will include your full peer review and any attached files.

Reviewer #1: No

---

## [Editor Report · Acceptance letter]

14 May 2020

PONE-D-19-29605R2 

The 4 Youth by Youth HIV Self-Testing Crowdsourcing Contest: A Qualitative Evaluation 

Dear Dr. Iwelunmor:

I am pleased to inform you that your manuscript has been deemed suitable for publication in PLOS ONE. Congratulations! Your manuscript is now with our production department. 

With kind regards,

on behalf of

Dr. Joseph K.B. Matovu 

Academic Editor

PLOS ONE